# Fluorescent detection of PARP activity in unfixed tissue

**Soumaya Belhadj**[1], **Andreas Rentsch**[2], **Frank Schwede**[2], **François Paquet-Durand**[1]*

**1** Cell Death Mechanism Group, Institute for Ophthalmic Research, Eberhard-Karls-Universität Tübingen, Tübingen, Germany, **2** Biolog Life Science Institute GmbH & Co. KG, Bremen, Germany

* francois.paquet-durand@klinikum.uni-tuebingen.de

**Data Availability Statement:** All relevant data are within the manuscript.

**Funding:** This research was funded by grants from the European Union (transMed; H2020-MSCA-765441), the Charlotte and Tistou Kerstan

## Abstract

Poly-ADP-ribose-polymerase (PARP) relates to a family of enzymes that can detect DNA breaks and initiate DNA repair. While this activity is generally seen as promoting cell survival, PARP enzymes are also known to be involved in cell death in numerous pathologies, including in inherited retinal degeneration. This ambiguous role of PARP makes it attractive to have a simple and fast enzyme activity assay, that allows resolving its enzymatic activity *in situ*, in individual cells, within complex tissues. A previously published two-step PARP activity assay uses biotinylated $NAD^+$ and streptavidin labelling for this purpose. Here, we used the fluorescent $NAD^+$ analogues ε-$NAD^+$ and 6-Fluo-10-$NAD^+$ to assess PARP activity directly on unfixed tissue sections obtained from wild-type and retinal degeneration-1 (*rd1*) mutant retina. In standard UV microscopy ε-$NAD^+$ incubation did not reveal PARP specific signal. In contrast, 6-Fluo-10-$NAD^+$ resulted in reliable detection of *in situ* PARP activity in *rd1* retina, especially in the degenerating photoreceptor cells. When the 6-Fluo-10-$NAD^+$ based PARP activity assay was performed in the presence of the PARP specific inhibitor olaparib, the activity signal was completely abolished, attesting to the specificity of the assay. The incubation of live organotypic retinal explant cultures with 6-Fluo-10-$NAD^+$, did not produce PARP specific signal, indicating that the fluorescent marker may not be sufficiently membrane-permeable to label living cells. In summary, we present a new, rapid, and simple to use fluorescence assay for the cellular resolution of PARP activity on unfixed tissue, for instance in complex neuronal tissues such as the retina.

## Introduction

Poly (ADP-ribose) polymerase (PARP) is a family of enzymes involved in numerous cellular processes [1, 2]. An important function of PARP is to detect DNA damage and to activate the enzymatic machinery involved in the DNA repair process. To this effect, PARP uses $NAD^+$ as a substrate to build poly-ADP ribose polymers (PAR) [3]. While this activity of PARP is generally seen as beneficial and cell-survival promoting, paradoxically, PARP enzymes are also known to be involved in cell death [4]. Notably, a specific type of caspase-independent programmed cell death, called PARthanatos, appears to be triggered by the accumulation of PAR polymers [5, 6]. These polymers may translocate to the mitochondria to cause the release of

Foundation, and the German research council (DFG; PA1751/8-1, 10-1). The funders provided support in the form of salaries for authors SB and FPD, but did not have any additional role in the study design, data collection and analysis, decision to publish, or preparation of the manuscript. Biolog Life Science Institute GmbH & Co. KG provided support in the form of salary for authors AR and FS, but had no role in study design, data collection and analysis, decision to publish, or preparation of the manuscript. The specific roles of these authors are articulated in the 'author contributions' section.

**Competing interests:** The authors have read the journal's policy and have the following competing interests: AR and FS are paid employees of Biolog Life Science Institute GmbH & Co. KG. This does not alter our adherence to PLOS ONE policies on sharing data and materials. The authors would like to declare the following patent/marketed product associated with this research: NAD+ analogues.

apoptosis-inducing factor (AIF) and its subsequent translocation to the nucleus, where AIF induces DNA fragmentation. PARthanatos is caspase-independent, as caspases would cleave and inactivate PARP [7]. Since PARP remains intact and active during the process, PAR synthesis is strongly increased, leading to the depletion of $NAD^+$ and ATP, further precipitating cell death [8]. This phenomenon may be relevant in a situation where PARP and its functional antagonist poly-ADP-ribose-glycohydrolase (PARG) [9] are activated simultaneously. While PARP can in principle self-regulate its activity via inhibitory auto-PARylation, the removal of this self-inhibition by PARG allows for continued and unabated activity of PARP.

PARP overactivation associated with AIF release has been shown to be involved in acute neuronal and myocardial cell death following ischemia reperfusion [10]. PARP is also known to have a role in various inflammatory disorders. For example, PARP-1 deficient mice, or mice that have been treated with PARP inhibitors, are resistant to various types of inflammation, including streptozotocin-induced diabetes and lipopolysaccharide-induced sceptic shock [10]. Moreover, in January 2018, olaparib (trade name Lynparza) became the first PARP inhibitor to be approved by the FDA for the treatment of metastatic breast cancer [11]. Altogether, these findings suggest that while PARP is already a target for therapies in oncology, it may also be of interest for therapy development in inflammatory, cardiac, and neurological disorders, including such that affect the neuroretina. In this context it then becomes highly interesting to be able to study PARP activity in an easy and rapid assay that enables the detection in individual cells. Previously, we have adapted and established such an *in situ* PARP activity assay, using the retina as a model for neurodegeneration [12, 13].

The retina is affected by a group of genetic diseases collectively referred to as inherited retinal degeneration (IRD) and characterized by a progressive degeneration of photoreceptors, leading to vision impairment and ultimately blindness. The pathological mechanisms behind IRD are still unresolved and effective therapy is lacking. The *rd1* mouse is to date one of the best characterized animal models for IRD. This animal suffers from a mutation in the beta sub-unit of the rod photoreceptor *Pde6* gene, producing a non-functional phosphodiesterase-6 (PDE6) and leading to an accumulation of cGMP [14]. Various studies [13, 15, 16] showed that inherited photoreceptor cell death involves PARP activity. Indeed, PARP activity, as well as its product PAR, are dramatically increased in the outer nuclear layer (ONL) of *rd1* mice at postnatal day (PN) 11, a time-point which corresponds to the onset of cell death in this animal model. Even further, PARP inhibition, *in vitro* and *in vivo*, resulted in increased photoreceptor survival [13, 17].

In the present study, we tested three different analogues of the PARP substrate $NAD^+$ to assess and compare their capacity to detect PARP activity in retinal tissue sections derived from wild-type and *rd1* mice. While our previously established *in situ* PARP activity assay required two separate steps for positive detection, the use of a fluorescent $NAD^+$ analogue allowed the detection of PARP activity in a single-step assay, with essentially the same specificity and detection rate, and ease of use in standard microscopic equipment as in the two-step assay. This single-step procedure is faster and easier to perform and would, in principle, also lend itself to *in vivo* biomarker development.

## Materials & methods

### Animals

C3H *rd1* and wild type (WT) mice were used [14]. Animals were housed under standard white cyclic lighting, had free access to food and water, and were used irrespective of gender. Animal protocols compliant with §4 of the German law of animal protection were reviewed and

approved by the Tübingen University committee on animal protection (*Einrichtung für Tierschutz, Tierärztlichen Dienst und Labortierkunde*; Registration No. AK02/19M).

## Histology

For histology, two different types of retinal tissue preparations were collected: 1) Retinal tissue sections fixed with 4% paraformaldehyde (PFA), to be used for PAR staining, and 2) retinal tissue that was directly subjected to cryosectioning without fixation, to be used for PARP activity assay.

**Fixed sections (for PAR DAB staining).** Animals were sacrificed in the morning (10–11 am), their eyes enucleated and fixed in PFA in 0.1 M phosphate buffer (pH 7.4) for 45 min at 4˚C. After fixation, tissues were washed for 10 min in PBS. For cryoprotection, they were incubated in 10% sucrose solution for 10 min, 20% sucrose solution for 20 min, and 30% sucrose solution for at least 30 min. The retinas were frozen in Tissue-Tek O.C.T. compound (Sakura Finetek Europe, Alphen aan den Rijn, Netherlands)-filled boxes. 12 μm tissue sections were prepared on a Thermo Scientific NX50 microtome (Thermo Scientific, Waltham, MA), thaw-mounted onto Superfrost Plus object slides (R. Langenbrinck, Emmendingen, Germany). Tissue sections were stored at -20˚C.

**Unfixed sections (for PARP in situ activity assay).** Animals were sacrificed in the morning (10-11am), their eyes enucleated and snap frozen on liquid nitrogen. Then, the eyes were embedded in Tissue-Tek (Sakura Finetek). Sagittal 12 μm sections were obtained as above and stored at -20˚C.

## PARP *in situ* activity assays

For the detection of PARP activity *in situ*, we used unfixed sections. For the PARP activity assay using biotinylated $NAD^+$, the reaction mixture (10 mM $MgCl_2$, 1 mM dithiothreitol, 5 μm 6-Biotin-17-$NAD^+$ (Biolog, Bremen, Germany, Cat. Nr.: N 012) in 100 mM Tris buffer with 0.2% Triton X100, pH 8.0) was applied to the sections for 2 h 30 min at 37˚C. After three 5 min washes in PBS, incorporated biotin was detected by avidin-Alexa fluor488 conjugate (Invitrogen, Carlsbad, CA, dilution 1:800 in PBS, incubation 1 h at room temperature). After three 5 min washes in phosphate buffered saline (PBS), the sections were mounted in Vectashield (Vector, Burlingame, CA). For controls, the biotinylated-$NAD^+$ was omitted from the reaction mixture, resulting in absence of detectable reaction product.

For the PARP activity assay using fluorescent $NAD^+$, the reaction mixture (10 mM $MgCl_2$, 1 mM dithiothreitol, 50 μM 6-Fluo-10-$NAD^+$ (Biolog, Cat. Nr.: N 023) or 50 μM ε-$NAD^+$ (Biolog, Cat. Nr.: N 024), in 100 mM Tris buffer with 0.2% Triton X100, pH 8.0) was applied to the sections for 2 h 30 min at 37˚C. After three 5 min washes in PBS the sections were mounted in Vectashield (Vector). For controls, the fluorescent-$NAD^+$ was omitted from the reaction mixture, resulting in absence of detectable reaction product.

## PAR DAB staining

For the detection of PAR DAB staining, we used fixed sections. 3,3′-diaminobenzidine (DAB) staining commenced with quenching of endogenous peroxidase activity using 40% MeOH and 10% $H_2O_2$ in PBS with 0.3% Triton X-100 (PBST) and 20 min incubation. The sections were further incubated with 10% normal goat serum (NGS) in PBST for 30 min followed by anti-PAR antibody (1:200; Enzo Life Sciences, Farmingdale, New-York) incubation overnight, at 4˚C. Incubation with the biotinylated secondary antibody (1:150, Vector; in 5% NGS in PBST) for 1 h was followed by application of Vector ABC-Kit (Vector Laboratories, solution A and solution B in PBS, 1:150 each) for 1 h. DAB staining solution (0.05 mg/ml $NH_4Cl$, 200 mg/ml

glucose, 0.8 mg/ml nickel ammonium sulphate, 1 mg/ml DAB, 0.1 vol. % glucose oxidase in phosphate buffer) was applied evenly, incubated for exactly 2 min and immediately rinsed with phosphate buffer to stop the reaction. The sections were mounted in Aquatex (Merck, Darmstadt, Germany).

### Inhibition of PARP activity using olaparib

The tissue sections were pre-incubated with olaparib (Biomol, Hamburg, Germany, 100 nM in PARP reaction buffer) for 30 min. The PARP activity assay was then conducted as described above with olaparib added to the reaction mixture at a concentration of 100 nM. Olaparib was omitted from the positive control.

### Testing fluorescent NAD$^+$ on live retina

At post-natal (P) day 5 *rd1* animals were killed and the eyes rapidly enucleated in an aseptic environment. The entire eyes were incubated in R16 serum-free culture medium (Gibco, Carlsbad, CA), with 0.12% proteinase K (MP Biomedicals, Illkirch-Grafenstaden, France), at 37˚C for 15 min, to allow preparation of retinal cultures with retinal pigment epithelium (RPE) attached. The proteinase K was inactivated with 10% FCS (Gibco) in R16 medium, and thereafter the eyes were dissected aseptically in a Petri dish containing fresh R16 medium. The anterior segment, lens, vitreous, sclera, and choroid were carefully removed by fine scissors, and the retina was cut perpendicular to its edges, giving a cloverleaf-like shape. Subsequently, the retina was transferred to a culture dish filter insert (COSTAR, NY) with the RPE layer facing the membrane. The insert was put into a six-well culture plate and incubated in R16 medium with supplements [18] at 37˚C. The full volume of nutrient medium, 1 ml per dish, was replaced with fresh R16 medium every second day.

At P11, the cultures were incubated for 24 h with 6-Fluo-10-NAD$^+$ at a concentration of 100 μM. At P12, the cultures were stopped by 45 min fixation in 4% PFA. This was followed by graded sucrose cryoprotection, embedding, and collecting of 12-μm-thick retinal cross-sections on a Thermo Scientific NX50 cryotome.

### Microscopy, cell counting, and statistical analysis

Light and fluorescence microscopy were performed at room temperature on an Axio Imager Z.1 ApoTome Microscope, equipped with a Zeiss Axiocam MRm digital camera (for more details about the channels used and fluorescence properties of the probes see Table 1). Images were captured using the Zeiss Zen software; representative pictures were taken from central areas of the retina using a 20x / 0.8 Zeiss Plan-APOCHROMAT objective. For quantifications, pictures were captured on three entire sagittal sections from at least three different animals. The average area occupied by a photoreceptor cell (*i.e.* cell size) was determined by counting 4′,6-diamidino-2-phenylindole (DAPI) stained nuclei in nine different areas of the retina. The total number of photoreceptor cells was estimated by dividing the outer nuclear layer (ONL) area by this average cell size. The number of positively labelled cells in the ONL was counted manually. Errors in graphs and text are given as standard deviation (STD).

**Table 1. Fluorescence of NAD$^+$ analogues and microscope filters.** Excitation (exc.) and emission (em.) characteristics of the two fluorescent probes and of the microscope filter sets used to visualize them.

| NAD$^+$ analogue | exc. max. / em. max. (nm) | microscope chan. | exc. filter / em. filter (nm) |
|---|---|---|---|
| 6-Fluo-10-NAD$^+$ | 494 / 517 | AF488 | 450–490 /500-550 |
| ε-NAD$^+$ | 300/410 | DAPI | 335–383 /420-470 |

For statistical comparisons, the unpaired Student t-test (PAR staining) and a two-way ANOVA with multiple comparison analysis (PARP activity assay) as implemented in Prism 8 for Windows (GraphPad Software, San Diego, CA) were employed. Adobe Photoshop (CS5A-dobe Systems Incorporated, San Jose, CA) was used for image processing and figure preparation.

## Results

### Using NAD$^+$ analogues to probe for *in situ* PARP activity

To detect the activity of PARP in unfixed tissue sections, we used three different analogues of NAD$^+$ that were differently substituted at the N$^6$ position of the molecule (Fig 1). Of these three analogues ε-NAD$^+$ and 6-Fluo-10-NAD$^+$ exhibited fluorescence properties (*cf.* Table 1), while the biotinylated 6-Biotin-17-NAD$^+$ was not by itself fluorescent.

The current assay to evaluate PARP activity *ex vivo* on tissue sections is using biotinylated NAD$^+$, which in a second step is recognized by fluorescently labelled streptavidin [12]. When this two-step assay was performed on *rd1* P11 retinal sections, a large number of positive cells was seen in the ONL, while essentially no positive cells were seen in the WT condition at the same age (Fig 2A).

The aim of the present study was to develop a faster and easier to use single-step PARP activity assay, that might potentially be suitable also for *in vivo* application. The two-step assay would indeed not be applicable *in vivo*. To this end, the two fluorescent analogues of NAD$^+$ were tested on *ex vivo* retinal tissue for their capacity to reliably detect PARP activity using standard microscopic equipment.

**Fig 1. Molecular structures of NAD$^+$-analogues used in this study.** The fluorescent properties of the NAD$^+$ analogues employed here are established via the additional etheno-function bridging the N$^6$- and 1-position (ε-NAD$^+$) or via the fluorescent dye fluorescein attached via a spacer of 10 bond lengths at the N$^6$-position (6-Fluo-10-NAD$^+$). 6-Biotin-17-NAD$^+$ is not fluorescent itself, but features a biotin moiety, which is also attached to the N$^6$-position (via a spacer of 17 bond lengths) and can be addressed by a fluorescent dye such as avidin-Alexa fluor488.

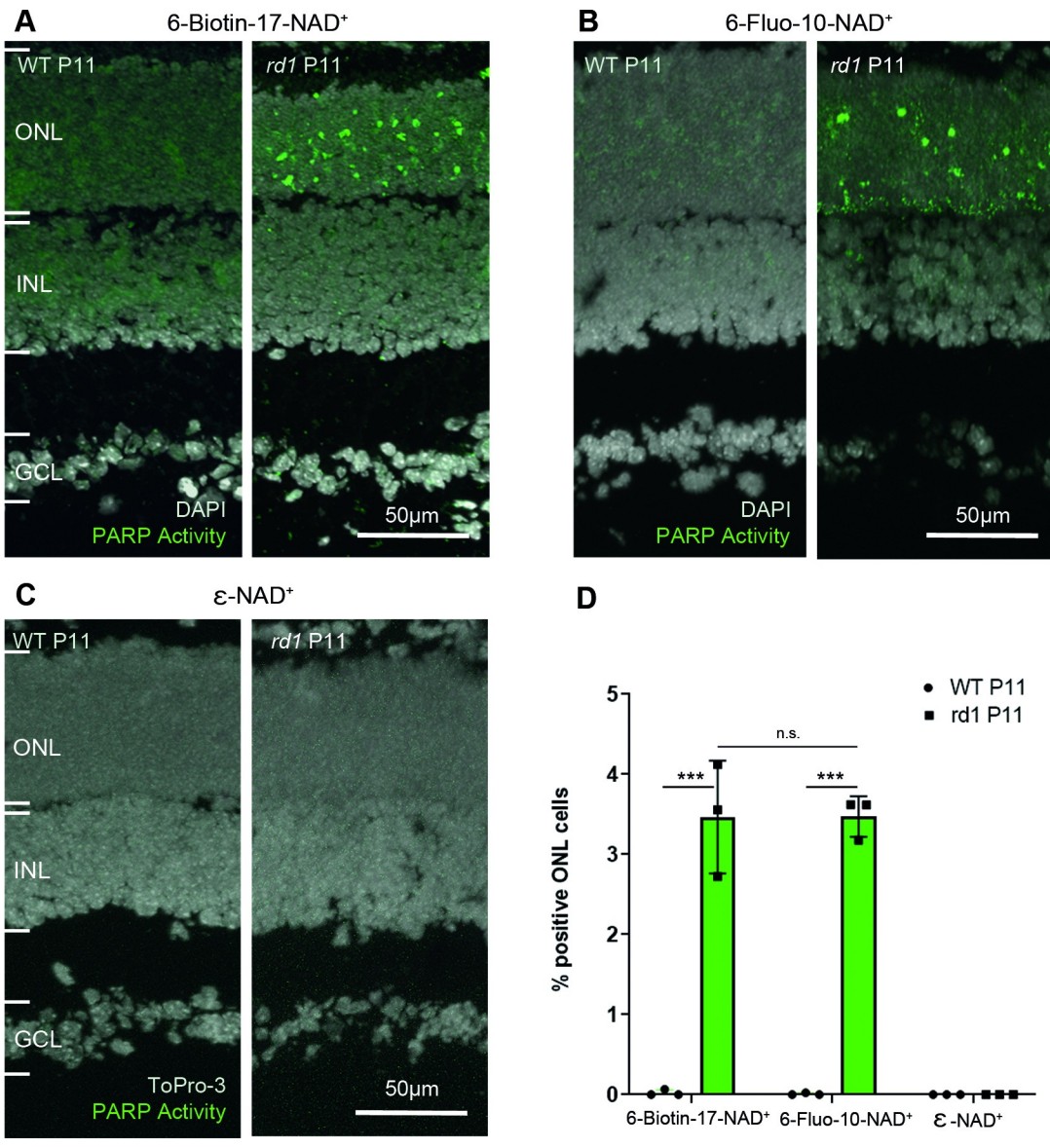

**Fig 2. PARP activity detection with three different NAD⁺ analogues, in wild-type and *rd1* retina.** Retinal tissue sections derived from either wild-type (WT) or *rd1* animals, at post-natal (P) day 11, were incubated with different NAD⁺ analogues. (**A**) In the two-step assay, employing 6-Biotin-17-NAD⁺, PARP activity positive cells were rarely seen in the WT retina but readily detected in the *rd1* outer nuclear layer (ONL). (**B**) The single-step assay with 6-Fluo-10-NAD⁺ detected similar numbers of PARP activity positive cells in the *rd1* ONL. (**C**) No PARP activity was observable with the assay employing ε-NAD⁺ and a standard UV band-pass filter intended for DAPI visualization. (**D**) Quantification of PARP activity positive cells in WT and *rd1* ONL in assays using the three different NAD⁺ analogues. DAPI was used as nuclear counterstaining in A and B; To-Pro-3 was used in C. PARP assays performed on tissue sections derived from at least three different WT and *rd1* animals (n = 3); error bars indicate STD; statistical analysis: two-way ANOVA with multiple comparison. INL = inner nuclear layer, GCL = ganglion cell layer.

When 6-Fluo-10-NAD⁺ was used as a substrate for PARP, a high amount of PARP positive cells was observed in the ONL of *rd1* P11 sections. Some signal was also seen at the lower border of the ONL, close to the outer plexiform layer (OPL) (Fig 2B). In contrast, ε-NAD⁺ did not reveal any PARP specific signal, neither in *rd1* tissue sections, nor in WT ones, when visualized with a standard UV band-pass filter for DAPI (Fig 2C, *cf*. Table 1).

When the single-step 6-Fluo-10-NAD$^+$ based assay was compared to the two-step 6-Biotin-17-NAD$^+$ assay, no significant difference was seen in the rate of PARP activity detection. In the *rd1* mouse model ONL at P11, both probes detected on average 3.47 ± 0.25 STD and 3.46 ± 0.70 STD PARP activity positive cells, respectively (Fig 2D).

To validate the results obtained for PARP activity with an independent method, we used an immunostaining for PAR to detect the cellular products of PARP activity and to thereby assess PARP activity indirectly. While some positive cells were observed in the ONL of the *rd1* mouse model at P11 (Fig 3B), none were seen in the WT condition at the same age (Fig 3A). In average, 0.99 ± 0.08 STD PAR positive cells were detected in the ONL of the *rd1* mouse model at P11 (Fig 3C).

### Inhibition of PARP activity: Effect on detection rate

To assess the specificity of the newly set up activity assay, the PARP inhibitor olaparib was used. Olaparib is inhibiting PARP1, PARP2, and PARP3 and is used as a therapy for cancer in people with hereditary BRCA1 or BRCA2 mutations, which include some ovarian, breast and prostate cancers [19]. This inhibitor has multiple modes of action. First, it competes with the binding of NAD$^+$, thus inhibiting PARylation. Additionally, olaparib traps PARP1 and PARP2 on the DNA strand, therefore interfering with DNA damage repair [11]. It has been reported that the IC$_{50}$ of olaparib for PARP1 is 5 nM, while it is 1 nM for PARP2 [20].

In the positive control condition, for which the PARP activity assay was conducted normally on *rd1* P11 retinal sections, a high amount of PARP positive cells was observed in the ONL (Fig 4A and 4C). This signal was extinguished when the activity assay was carried out on retinal sections from the same animal model, at the same age, with olaparib present in the reaction mixture at a concentration of 100 nM (Fig 4B and 4D). These results suggest that both activity assays are specific of PARP activity.

### Testing NAD$^+$ analogues on live retina *in vitro*

To assess whether 6-Fluo-10-NAD$^+$ could be used to assess PARP activity in live tissue, with intact cell membranes, organotypic retinal explant cultures [21–23] were treated with this probe at a concentration of 100 μM for 24h. No significant difference was observed between the negative control condition (not treated with 6-Fluo-10-NAD$^+$) and the treated condition, and no PARP specific signal could be seen (Fig 5A and 5B). These negative results indicate that 6-Fluo-10-NAD$^+$ may not be able to penetrate the membranes of living cells.

## Discussion

The detection and quantification of cellular PARP activity is highly desirable in the context of many diseases, including in neurodegenerative diseases of the retina. Here, we identified a fluorescent analogue of NAD$^+$ that enables the rapid *in situ* detection of PARP activity in a simple, single-step assay procedure. This assay could potentially facilitate the development of drugs targeting PARP, and, because of PARPs connection to cell death, could be used to improve our understanding of cell death mechanisms. Moreover, PARP activity detection could also serve as a surrogate biomarker for the study and diagnosis of degenerative diseases.

### Methods for the detection of PARP activity

PARP activity can be assessed by a number of commercially available assay kits, which detect PARP activity in tissue lysates or cell cultures [24]. The drawback of these assays is that they do not offer cellular resolution, something that is highly desirable in the context of studies into

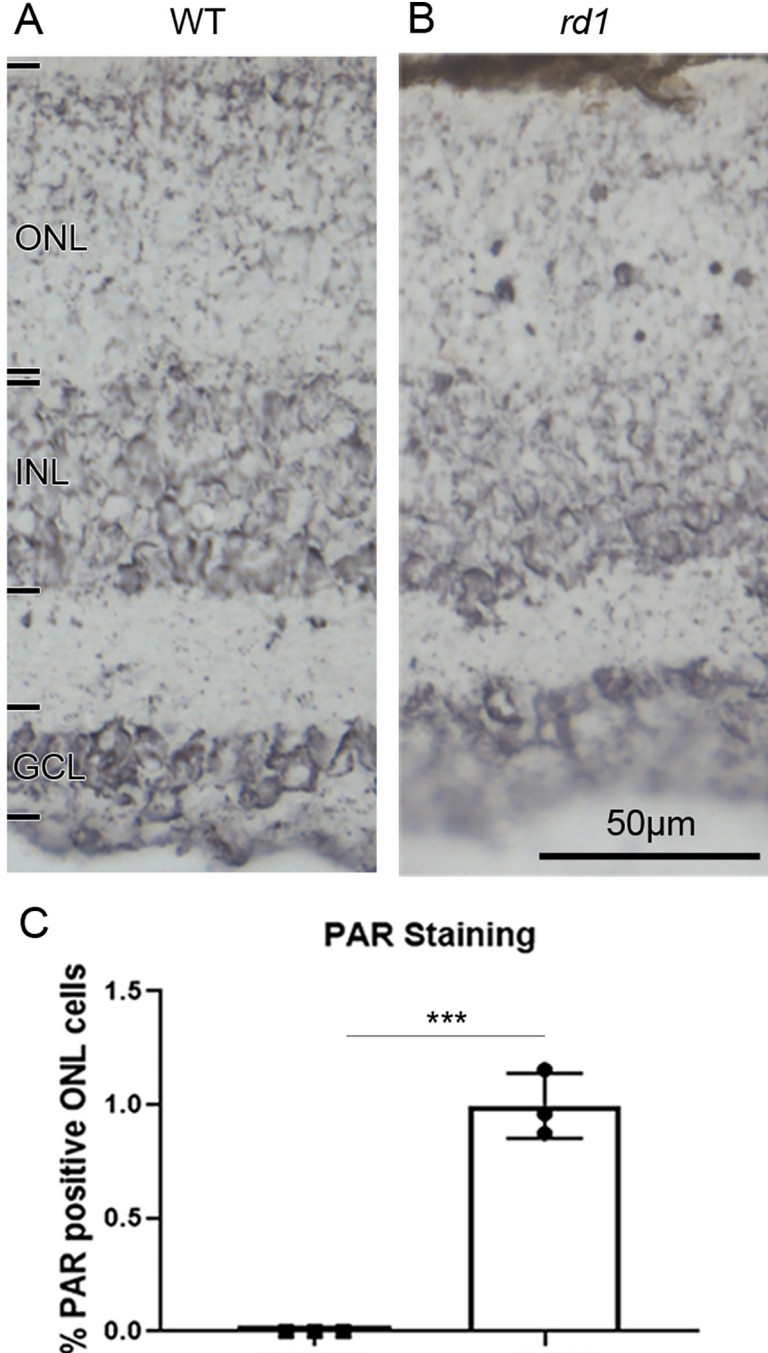

**Fig 3. Immunostaining for poly-ADP-ribosylation in wild-type and *rd1* retina.** To indirectly assess PARP activity an immunostaining for poly-ADP-ribosylation (PAR) was employed. While post-natal (P) day 11 wild-type (WT) outer nuclear layer (ONL) is essentially devoid of nuclear PAR staining (**A**), in the *rd1* situation (**B**) numerous photoreceptor nuclei show a marked PAR label. (**C**) Quantification of PAR positive nuclei in the ONL, with n = 3. error bars indicate STD; statistical analysis: Student´s t-test. INL = inner nuclear layer, GCL = ganglion cell layer. Images representative for PAR immunostaining performed on tissue sections derived from at least three different WT and *rd1* animals.

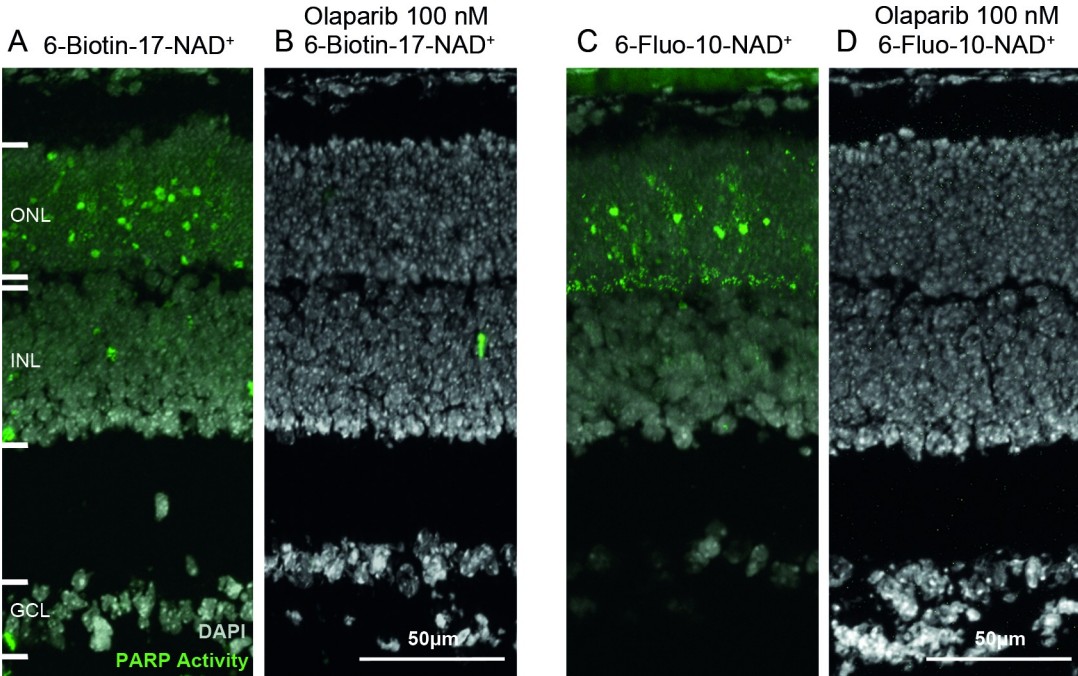

**Fig 4. The PARP inhibitor olaparib abolishes PARP activity signals.** Retinal sections derived from post-natal (P) day 11 *rd1* animals were incubated with NAD$^+$ analogues, together with the selective PARP inhibitor olaparib. In both the two-step assay with 6-Biotin-17-NAD$^+$ (**A**, **B**) and the single-step assay using 6-Fluo-10-NAD$^+$ (**C**, **D**), olaparib, at 100 nM, completely abolished positive PARP activity detection. DAPI was used as nuclear counterstaining. ONL = outer nuclear layer, INL = inner nuclear layer, GCL = ganglion cell layer. Images representative for assays performed on tissue sections derived for at least three different *rd1* animals per assay condition.

specific disease pathomechanisms, for instance. Another way to assess PARP activity in whole tissue lysates is to perform an activity blot. Using whole tissue samples, the technique combines western blotting followed by a protein renaturation and activity assay. A labelling will be seen in protein bands with molecular weights corresponding to PARP enzymes [25]. While this technique may allow attributing activity to specific protein bands or isoforms, again, the

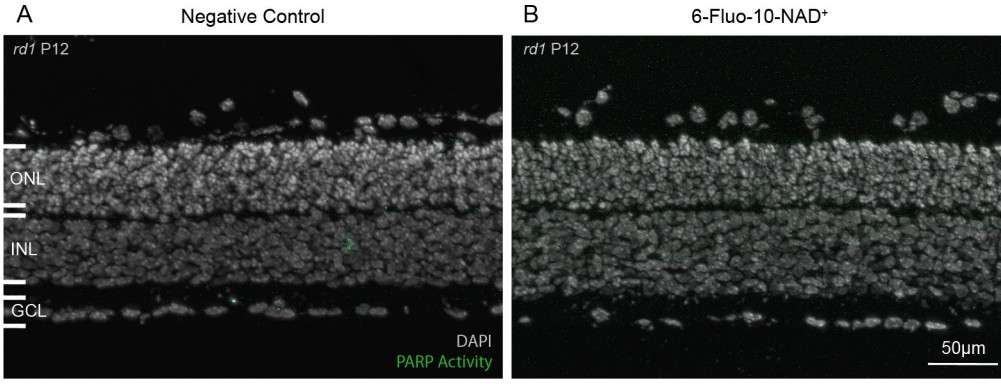

**Fig 5. The 6-Fluo-10-NAD$^+$ analogue does not allow live tissue detection of PARP activity.** Live, organotypic retinal explant cultures were incubated with 100 μM 6-Fluo-10-NAD$^+$ from P11 to P12 (24h). No fluorescent signal was observed within the retina, indicating that the NAD$^+$ analogue may not have penetrated live cells. DAPI was used as nuclear counterstaining. ONL = outer nuclear layer, INL = inner nuclear layer, GCL = ganglion cell layer. Images representative for results obtained from 5 independent retinal explant cultures.

drawback is that it does not offer cellular resolution. PARP activity can also be assessed indirectly at the cellular level by immunohistology with antibodies against PAR. The staining can be performed on cell cultures, sectioned tissue, or after western blotting [13]. The disadvantage of this method is that PARP activity and PAR product accumulation may not necessarily correspond. PAR may, for instance, be hydrolyzed secondarily by the specific glycohydrolase PARG, potentially reducing detection sensitivity.

## Using NAD⁺ analogues for the *in situ* detection of PARP activity

The above-mentioned methods do not allow for the detection of PARP activity within individual cells. This, however, is highly relevant in complex tissues, such as when they occur in the central nervous system where only some of the many different cell types may be affected by a disease condition or where only a specific cell type may be amenable to PARP inhibition treatment. As mentioned previously, PARP inhibition is a therapeutic strategy for some cancers [11]. To help the diagnosis with PARP inhibitor therapy, Shuhendler *et al*. developed a novel $NAD^+$ analogue radioactive probe ([$^{18}$F]-SuPAR) for noninvasive imaging of PARP activity using positron emission tomography. The probe was tested in breast and cervical cancer xenograft mouse models [26]. Recently, Wigle *et al*. developed a set of $NAD^+$ competitors using FRET/BRET for *in vitro* and cellular high-throughput biophysical assays to study PARP activity and inhibition [27]. In this study, to assess PARP activity *in situ*, a new single-step assay, based on the use of the fluorescent $NAD^+$ analogue 6-Fluo-10-$NAD^+$, was set up. This new assay had a similar detection rate compared to the established two-step assay using biotinylated $NAD^+$ (6-Biotin-17-$NAD^+$). However, the sensitivity appeared to be higher for the two-step activity assay using biotinylated $NAD^+$, since the concentration of 6-Fluo-10-$NAD^+$ (50 μM) used to perform the assay was ten times higher than the one of 6-Biotin-17-$NAD^+$ (5 μM). This can be explained by the specific amplification of the signal provided by the fluorescently labelled avidin used in the second detection step.

When the activity assay was performed with ε-NAD, and with the limitations of standard microscopic equipment and filter sets, no PARP specific signal could be observed. This is, to some extent, in contrast to a previous study, which reported the use of ε-$NAD^+$ to assess PARP activity in cultured cells lines and rat brain slices [28]. However, the $NAD^+$ analogue concentration used by Davis *et al*. was 400 μM (8X compared to the concentration used here) and even then, the PARP-specific signal was detectable only after incubation with an anti-ethenoadenosine antibody as a second step, to detect ε-$NAD^+$ containing PAR polymers. Combined with the fact that 300 nm UV excitation is required, these limitations make ε-$NAD^+$ appear poorly suited for *in situ* detection assays, especially so if live cell detection was required, as the high energy UV irradiation would like result in significant cell damage.

## Specificity of *in situ* PARP activity detection

$NAD^+$ is very widely used by enzymes as a cofactor or substrate. This fact may raise questions about the specificity of the assay as the fluorescence observed could originate from the activity of other enzymes as well. To address this potential issue, we used PAR immunostaining as an indirect validation of the activity assay. While the PARP activity assay detected approximately 3.5% positive cells on average, in the *rd1* mouse model at postnatal day 11, the PAR immunostaining detected around 1% positive cells on average in the same model, at the same age. This corresponds very well with previous observations on the relation of PARP activity positive to PAR positive cells [13]. Nevertheless, this difference between PARP activity and PAR detection might suggest that some cells detected as PARP positive were, in fact, positive for another enzyme that used $NAD^+$. This could, for instance, be the case for the signal observed at the

ONL to OPL border when the assay was performed with 6-Fluo-10-NAD$^+$. This OPL signal was not revealed by the PAR immunostaining. Several reasons could explain this discrepancy: 1) the PAR polymers produced in the photoreceptor synapses may be not long enough to be detected by the PAR antibody. 2) the activity of an enzyme hydrolyzing PAR polymers in the synapse (such as PARG) may degrade the polymers at this location only. 3) the PAR antibody may not reach into the synapse as well as in the nucleus or the cytoplasm. 4) the signal may have been produced by an isoform of PARP present in the synapses only. On the other hand, when the single-step assay using 6-Fluo-10-NAD$^+$ was performed in the presence of PARP inhibitor olaparib, the staining produced in both photoreceptor nuclei and along the OPL disappeared. Altogether, these findings strongly suggest that the staining produced by 6-Fluo-10-NAD$^+$ incubation was indeed specific for the activity of PARP.

## Tissue penetration and permeability of 6-Fluo-10-NAD$^+$

Once the assay and its specificity for PARP was established, we wanted to see whether the single-step PARP activity detection could also be applied to live cells and tissues, and potentially even *in vivo*. In the retina such a method could potentially enable a direct, non-invasive detection of PARP activity *in vivo*, with single-cell resolution, using techniques such adaptive optics scanning laser ophthalmoscopy (AO-SLO) [29–31].

As opposed to what we had observed in unfixed retinal tissue sections, in living organotypic retinal explant cultures incubation 6-Fluo-10-NAD$^+$ did not reveal any PARP specific signal. This could indicate that the probe was either not able to penetrate into the tissue or could not reach across the intact cell membranes of live cells. Such a lack of permeability may be explained by the negative net charge of the 6-Fluo-10-NAD$^+$ molecule (Fig 1). It is likely that this property limits any *in vivo* application of the fluorescent probe. A possible solution might be to change the molecule in way that would reduce its net charge. This, however, could also affect its capacity to serve as substrate for PARP. An alternative solution might be to encapsulate the molecule in a delivery system such as liposomes or nanoparticles [32] that would release the molecule only intracellularly. In summary, while the 6-Fluo-10-NAD$^+$ compound appears well suited for PARP activity detection on unfixed *ex vivo* tissue preparations, it cannot readily be used for such purpose in live tissue.

## Conclusion

In this study, we set up a new, simple to perform, one-step assay to assess PARP activity on unfixed tissue, using the NAD$^+$ analogue 6-Fluo-10-NAD$^+$. The apparent sensitivity of this single-step assay, as determined by the positive cell detection rate, is similar to that of the previously established two-step assay that used 6-Biotin-17-NAD$^+$. The specificity of the new PARP activity assay was confirmed on the one hand by comparison with PAR immunostaining and on the other hand by inhibition of PARP activity with olaparib. Altogether, the new PARP activity assay is well suited for *ex vivo* applications on unfixed, native tissue. Moreover, further development and combination with a suitable delivery system could potentially enable future use as an *in vivo* biomarker for PARP activity [33, 34].

## Acknowledgments

We thank Norman Rieger for excellent technical assistance and Mathias Seeliger, Timm Schubert, Valeria Marigo, Per Ekström, and John Groten for helpful discussions. We thank Ayse Sahaboglu for kindly providing olaparib, Yiyi Chen for help with figure making, and Torsten Strasser for help with statistical analysis.

## Author Contributions

**Data curation:** Soumaya Belhadj.

**Formal analysis:** Soumaya Belhadj.

**Funding acquisition:** François Paquet-Durand.

**Investigation:** Soumaya Belhadj.

**Methodology:** Soumaya Belhadj.

**Supervision:** François Paquet-Durand.

**Writing – original draft:** Soumaya Belhadj.

**Writing – review & editing:** Andreas Rentsch, Frank Schwede, François Paquet-Durand.

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
