## [Decision Letter · Decision Letter 0]

15 Jul 2020

PONE-D-20-16995

Fluorescent detection of PARP activity in unfixed tissue

PLOS ONE

Dear Dr. Belhadj,

Thank you for submitting your manuscript to PLOS ONE. After careful consideration, we feel that it has merit but does not fully meet PLOS ONE’s publication criteria as it currently stands. Therefore, we invite you to submit a revised version of the manuscript that addresses the points raised during the review process.

You will see that there is disagreement between the expert reviewers concerning the merit of this submission. At a minimum, you should fully cite and acknowledge alternative methods for labeling PARP activity in tissue, and immunocytochemistry to illustrate PAR staining should be performed *on the same sections* as those stained with 6-Fluo-10-NAD. Because there is an apparent conflict of interest involving two of the authors, the novelty  and relative value of your method should be explained.

We look forward to receiving your revised manuscript.

Kind regards,

Alfred S Lewin, Ph.D.

Academic Editor

PLOS ONE

Journal Requirements:

2. Thank you for including your competing interests statement; "I have read the journal's policy and the authors of this manuscript have the following competing interests: AR and FS are employees of Biolog Life Science Institute GmbH & Co. KG, which is selling NAD+ analogues."

We note that one or more of the authors are employed by a commercial company: Biolog Life Science Institute GmbH & Co. KG

Please respond by return email with an updated Funding Statement and Competing Interests Statement and we will change the online submission form on your behalf.

3. 

PLOS requires an ORCID iD for the corresponding author in Editorial Manager on papers submitted after December 6th, 2016. Please ensure that you have an ORCID iD and that it is validated in Editorial Manager. To do this, go to ‘Update my Information’ (in the upper left-hand corner of the main menu), and click on the Fetch/Validate link next to the ORCID field. This will take you to the ORCID site and allow you to create a new iD or authenticate a pre-existing iD in Editorial Manager. Please see the following video for instructions on linking an ORCID iD to your Editorial Manager account: https://www.youtube.com/watch?v=_xcclfuvtxQ

Reviewers' comments:

Reviewer's Responses to Questions

**Comments to the Author**

1. Is the manuscript technically sound, and do the data support the conclusions?

Reviewer #1: Yes

Reviewer #2: Yes

2. Has the statistical analysis been performed appropriately and rigorously? 

Reviewer #1: Yes

Reviewer #2: No

3. Have the authors made all data underlying the findings in their manuscript fully available?

Reviewer #1: Yes

Reviewer #2: Yes

4. Is the manuscript presented in an intelligible fashion and written in standard English?

Reviewer #1: Yes

Reviewer #2: Yes

5. Review Comments to the Author

Reviewer #1: The present study by Belhadj et al. describes a new assay for detection of PARP activity, which is easier to perform than previously reported two-step procedures. It requires only one step and is demonstrated to work on non-fixed tissue. In order to be useful for the scientific community, the used reagents need to be masde available, either by describing the procedures for their5 preparation and their analytical data or (I understand that two of the authors are employees of a company called Biology, which is specialized in selling nucleotide derivatives), if the company may hesitate to describe the synthesis, needs to be made commercially available. For the biotinylated derivative this seems to be the case (?, see p. 5, PARP in situ activity assays, also for the fluorescent probe, p.5, l. 111), but it would be good to state this clearly in the manuscript .

The fluorescent, as well as the biotinylated analog seem to work nicely in unfixed tissue samples, albeit all of them look rather cell impermeable. I could not detect any permeabilization step in preparation of the tissue samples, is this a specific property of the retinal samples? Also the antibody incubation works nicely, so permeability of the tissue is obviously enhanced. The authors should comment here.

Is it possible to determine the KD of the used probes to PARP? The conclusiuon says that the sensitivity of the one-step assay is somewhat lower (p.15, l 338), the introduction claims the sensitivity of the two assay systems to be essentially the same (p.10, l. 79). This should be homogenized. The point of decreased sensitivity is possibly not completely correct, as fluorescein is not an extremely bright and photostable dye (and as the authors state correctly, the antibody two-step protocol also offers the advantage of signal amplification), so sensitivity of detection is lower, but sensitivity in terms of interacting with PARP may still be very high (just the antibody format employing multiple dyes).

Is there an explanation for the different location of fluorescent cells in Figure 2 and 4? Panel a (biotin) and panel b look quite different. This should be explained.

The conclusion could still discuss options for further improvement of the current probes in terms of reaching permeability in live cell samples – which would represent another big advantage over the current two-step protocol.

Minor point: p.15, l. 311 says Ref. Paquet-Durand 2007, please update.

In conclusion, the present manuscript needs some minor clarification and addition but the contained data is valuable, well presented and should be published once the points raised above have been addressed.

Reviewer #2: The manuscript by Belhadj et al describes the implementation of a commercially available fluorophore substrate for PARP, provided by the company where the authors are employed. The work deals with testing 6-Fluo-10-NAD in ex vivo retinal sections, comparing to established epsilon NAD and a biotinylated NAD. While the authors show that the signal produced by 6-Fluo-10-NAD in ex vivo sections is similar to that using the biotinylated compound, the novelty of this work is limited. The authors fail to report on other published methods for ex vivo PARP activity labelling in tissues (doi.org/10.1021/acs.bioconjchem.9b00089). They also fail to fully demonstrate the proportion of PAR positive cells stained by 6-Fluo-10-NAD. While immunocytochemistry was performed to illustrate PAR staining, it was done on different sections than the ones stained by 6-Fluo-10-NAD. As previously demonstrated (doi.org/10.1021/acs.bioconjchem.9b00089), the same sections should have been stained with both 6-Fluo-10-NAD and anti-PAR antibody for analysis by immunofluorescence microscopy. Otherwise, specificity is not exhaustively shown as claimed. Considering that fixation was performed after staining, it is not clear why this couldn't have been done.

Additionally, the value of the method presented is unclear: After staining with 6-Fluo-10-NAD, the authors still fix the tissue prior to imaging. Why do we need to stain before fixation then? While there are claims that this approach preserves the fraction of PAR from PARG-mediated degradation, there is no demonstration that pre- versus post-fixation labelling has any impact on PAR compliment.

The paper states that a two-way ANOVA was used for PARP activity assay evaluation, but the need for two-way versus one-way with posthoc evaluation is unclear, and the validity of the analysis is uncertain.

The authors also state that this work may open avenues for detecting PARP activity in vivo. Again, this has already been done (doi.org/10.1021/acs.bioconjchem.9b00089).

Overall, this manuscript appears to be more of a white paper, describing the implementation of a commercial dye for an assay that is already established. The case for adopting a one-step versus two-step method is not justified, and the claimed specificity is not directly or robustly demonstrated. The work does not add new knowledge or capability to the PARP biochemistry field.

6. PLOS authors have the option to publish the peer review history of their article (what does this mean?). If published, this will include your full peer review and any attached files.

Reviewer #1: No

Reviewer #2: No

---

## [Author Response · Author response to Decision Letter 0]

3 Nov 2020

Reviewer 1: 

The present study by Belhadj et al. describes a new assay for detection of PARP activity, which is easier to perform than previously reported two-step procedures. It requires only one step and is demonstrated to work on non-fixed tissue. 

In order to be useful for the scientific community, the used reagents need to be made available, either by describing the procedures for their preparation and their analytical data or (I understand that two of the authors are employees of a company called Biology, which is specialized in selling nucleotide derivatives), if the company may hesitate to describe the synthesis, needs to be made commercially available. For the biotinylated derivative this seems to be the case (?, see p. 5, PARP in situ activity assays, also for the fluorescent probe, p.5, l. 111), but it would be good to state this clearly in the manuscript.

Response: We thank the reviewer for this suggestion. We note that the all the reagents used in the study are indeed commercially available. However, we may not have expressed this clearly enough. To improve clarity, we have rephrased it in the Materials & Methods and added their catalogue numbers (l. 107, 114, and 115)

The fluorescent, as well as the biotinylated analog seem to work nicely in unfixed tissue samples, albeit all of them look rather cell impermeable. I could not detect any permeabilization step in preparation of the tissue samples, is this a specific property of the retinal samples? Also the antibody incubation works nicely, so permeability of the tissue is obviously enhanced. The authors should comment here.

Response: There is indeed no separate permeabilization step. However, the PARP reaction mixture contains Triton X100 (l. 107 and 115). Regarding the immunostaining, PBS with 0,3% Triton X100 was used in several steps of the protocol: quenching, blocking and incubations with both primary and secondary antibodies (l. 122, 123 and 125). 

Is it possible to determine the KD of the used probes to PARP? The conclusion says that the sensitivity of the one-step assay is somewhat lower (p.15, l 338), the introduction claims the sensitivity of the two assay systems to be essentially the same (p.10, l. 79). This should be homogenized. The point of decreased sensitivity is possibly not completely correct, as fluorescein is not an extremely bright and photostable dye (and as the authors state correctly, the antibody two-step protocol also offers the advantage of signal amplification), so sensitivity of detection is lower, but sensitivity in terms of interacting with PARP may still be very high (just the antibody format employing multiple dyes).

Response: We have not determined the KD values for the different NAD+ analogues. In the original submission there was indeed a discrepancy between what we said about probe sensitivity in introduction and conclusion. The reason for this was exactly as described by the reviewer, the lack of an amplification step in the single-step assay may reduce the apparent sensitivity, even though the binding affinities of the two PARP probes may in fact be very similar. We have now rephrased this aspect and harmonized the statements made in introduction (l. 80-83), discussion (l. 296-303), and conclusion (l. 352-354). 

Is there an explanation for the different location of fluorescent cells in Figure 2 and 4? Panel a (biotin) and panel b look quite different. This should be explained. 

Response: In Figure 2 (A, B) and Figure 4 (A, C) the results of the different PARP activity assays are shown. We did not observe a systematically different localization of PARP activity staining between 6-Biotin-17-NAD+ and 6-Fluo-10-NAD+. Nevertheless, when compared to the PAR immunostaining, which essentially shows only a nuclear staining in the outer nuclear layer (ONL; Figure 3), the NAD+ based activity assays both show an additional staining of the photoreceptor synapses in the outer plexiform layer (OPL). Since the fluorescent labelling was completely abolished by the PARP inhibitor olaparib (Figure 4), both the nuclear as well as the synaptic label are very likely PARP specific. 

To account for the reviewer’s comment, we have changed the images shown in figure 4 to more accurately represent the staining obtained with the different NAD+ analogues. 

The conclusion could still discuss options for further improvement of the current probes in terms of reaching permeability in live cell samples – which would represent another big advantage over the current two-step protocol.

Response: We thank the reviewer for this suggestion. Unfortunately, it may be difficult to change the structure of the molecule without affecting its PARP substrate properties. We now present this difficulty in the discussion (l. 342-347) and suggest as alternative solution the encapsulation in a drug delivery system such as liposomes or nanoparticles. 

Minor point: p.15, l. 311 says Ref. Paquet-Durand 2007, please update.

Response: We thank the reviewer for pointing this out. The reference has been updated (l. 320). 

In conclusion, the present manuscript needs some minor clarification and addition but the contained data is valuable, well presented and should be published once the points raised above have been addressed.

Reviewer 2:

The manuscript by Belhadj et al describes the implementation of a commercially available fluorophore substrate for PARP, provided by the company where the authors are employed. 

Response: Indeed, two of the co-authors are working at Biolog, a company that, among other compounds, produces and sells NAD+ analogues. This was clearly stated in the conflicts of interest section and is also obvious from the author affiliations. The fact that the used NAD+ analogues are from the company Biolog was also specified in the materials and methods section. Nevertheless, to address the reviewer´s concern, this has now been highlighted even further by giving the respective catalog numbers. 

The work deals with testing 6-Fluo-10-NAD in ex vivo retinal sections, comparing to established epsilon NAD and a biotinylated NAD. While the authors show that the signal produced by 6-Fluo-10-NAD in ex vivo sections is similar to that using the biotinylated compound, the novelty of this work is limited. The authors fail to report on other published methods for ex vivo PARP activity labelling in tissues (doi.org/10.1021/acs.bioconjchem.9b00089). 

Response: We thank the reviewer for this suggestion. The suggested publication is now included in the discussion (l. 291-294).

They also fail to fully demonstrate the proportion of PAR positive cells stained by 6-Fluo-10-NAD. While immunocytochemistry was performed to illustrate PAR staining, it was done on different sections than the ones stained by 6-Fluo-10-NAD. As previously demonstrated (doi.org/10.1021/acs.bioconjchem.9b00089), the same sections should have been stained with both 6-Fluo-10-NAD and anti-PAR antibody for analysis by immunofluorescence microscopy. Otherwise, specificity is not exhaustively shown as claimed. Considering that fixation was performed after staining, it is not clear why this couldn't have been done.

Response: We thank the reviewer for this question. The fixation of tissue samples would destroy the activity of the PARP enzyme, hence, it is essential to use unfixed tissue for the PARP activity assay. On the other hand, the immunostaining for PAR necessitates PFA fixed tissue. This is why a co-localization of PARP activity assay and PAR immunostaining is not easily done. 

Nevertheless, in response to this comment, we attempted to perform the co-localization as requested, on the same sections, stained both with 6-Fluo-10-NAD+ (and, as additional control, also with 6-Biotin-17-NAD+) and anti-PAR antibody. To this end, we used unfixed tissue to first perform the PARP activity assay, then we fixed the tissue section with 4% PFA, and subsequently performed PAR immunostaining on that same section. As controls, we used the individual staining procedures performed separately on unfixed and fixed tissue, respectively. The results of these stainings are shown in the Figure 1 below, which we provide for review purposes only. 

In control situations, the PARP activity assay revealed numerous positive cells (Figure 1A). Likewise, the immunostaining for PAR revealed positive cells in the outer nuclear layer (ONL; Figure 1B). However, when PARP activity assay was performed first and then followed by fixation, quenching of endogenous peroxidase activity, PAR staining, and numerous washing steps, PARP activity label was no longer detectable (Figure 1C). Conversely, the PAR staining, when performed on unfixed sections, which were then fixed only after the PARP activity assay had been performed, displayed a strong background and no positive label (Figure 1D). 

Figure 1: Colocalization of PARP Activity and PAR DAB Immunostaining. In the control panel (left), the PARP activity assay was performed on unfixed tissue sections as described in the manuscript (A), while the PAR DAB staining was performed on fixed tissue sections (B). In the co-localization panel (right) (C, D), the PARP activity assay was performed first on unfixed tissue sections, then, these were fixed with 4% PFA, before performing the PAR DAB staining. Note that PARP activity was undetectable in the co-localization experiment, while PAR staining delivered a very high background without specific labeling. DAPI (grey) was used as nuclear counterstaining in A and C. ONL = outer nuclear layer, INL = inner nuclear layer, GCL = ganglion cell layer.

Additionally, the value of the method presented is unclear: After staining with 6-Fluo-10-NAD, the authors still fix the tissue prior to imaging. Why do we need to stain before fixation then? While there are claims that this approach preserves the fraction of PAR from PARG-mediated degradation, there is no demonstration that pre- versus post-fixation labelling has any impact on PAR compliment.

Response: Here, there seems to be a misunderstanding. The PARP activity assay is, in fact, performed on unfixed tissue sections, which are then directly mounted for microscopic imaging, without the need for fixation. 

Moreover, we did not make claims as to the effects of pre- or post-fixation on PAR immunodetection. However, judging from the co-localization study performed for this revision (see Figure 1 above), we may now say that post-fixation is not suitable for PAR immunodetection. 

The paper states that a two-way ANOVA was used for PARP activity assay evaluation, but the need for two-way versus one-way with posthoc evaluation is unclear, and the validity of the analysis is uncertain.

Response: We believe the 2-way ANOVA analysis is important because we are interested not only in the effects of the NAD+ analogues but also in the effects on the different groups (wt vs. rd1) and, more importantly, on the interactions between NAD+ analogues and groups. Nevertheless, we simplified the analysis by excluding the ε-NAD+ group (which was essentially 0) and confirmed the model’s perquisites for normality of the residuals. We have revised the corresponding part of the materials and methods section to better explain the statistical analysis (l. 168-170).

The authors also state that this work may open avenues for detecting PARP activity in vivo. Again, this has already been done (doi.org/10.1021/acs.bioconjchem.9b00089).

Response: We thank the reviewer for pointing us to this very interesting paper. In Shuhendler et al., the authors developed a novel radioactive probe for in vivo imaging of PARP activity using positron emission tomography. The probe was tested in cancer xenograft mouse models. 

As opposed to the above study, we are interested in developing a fluorescent probe that can be combined with light-based, single-cell imaging techniques. As such the probe should be detectable both with conventional fluorescence microscopy and possibly also with current non-invasive in vivo imaging techniques such as adaptive optics scanning laser ophthalmoscopy. We now briefly present this concept in the discussion of the manuscript (l.334-338)

A further important difference to the Shuhendler et al., technique is that our assay does not require radioactive label nor PET detection, making it simpler and cheaper to handle for most laboratories. 

Overall, this manuscript appears to be more of a white paper, describing the implementation of a commercial dye for an assay that is already established. The case for adopting a one-step versus two-step method is not justified, and the claimed specificity is not directly or robustly demonstrated. The work does not add new knowledge or capability to the PARP biochemistry field.

Response: We thank the reviewer for raising this point, but we obviously beg to differ. To the very best of our knowledge, the assay using 6-Fluo-10-NAD+ has not previously been established for single cell PARP activity detection. Moreover, the single-step assay is faster and easier to perform, and can be implemented with standard lab equipment, allowing for a more rapid adoption by other laboratories. In the revised manuscript, we have now tried to more clearly highlight the advantages of the new in situ PARP activity assay (l. 189-191; l.267-270; l.351-358).

---

## [Decision Letter · Decision Letter 1]

2 Dec 2020

PONE-D-20-16995R1

Fluorescent detection of PARP activity in unfixed tissue

PLOS ONE

Dear Dr. Belhadj,

Thank you for submitting your manuscript to PLOS ONE. After careful consideration, we feel that it has merit but does not fully meet PLOS ONE’s publication criteria as it currently stands. Therefore, we invite you to submit a revised version of the manuscript that addresses the points raised during the review process.

You have not met the reviewer's request to address the colocalization of 6Fluo10NAD+ with PAR.

We look forward to receiving your revised manuscript.

Kind regards,

Alfred S Lewin, Ph.D.

Academic Editor

PLOS ONE

Reviewers' comments:

Reviewer's Responses to Questions

**Comments to the Author**

1. If the authors have adequately addressed your comments raised in a previous round of review and you feel that this manuscript is now acceptable for publication, you may indicate that here to bypass the “Comments to the Author” section, enter your conflict of interest statement in the “Confidential to Editor” section, and submit your "Accept" recommendation.

Reviewer #2: (No Response)

2. Is the manuscript technically sound, and do the data support the conclusions?

Reviewer #2: Partly

3. Has the statistical analysis been performed appropriately and rigorously? 

Reviewer #2: Yes

4. Have the authors made all data underlying the findings in their manuscript fully available?

Reviewer #2: Yes

5. Is the manuscript presented in an intelligible fashion and written in standard English?

Reviewer #2: Yes

6. Review Comments to the Author

Reviewer #2: I appreciate the care the authors have taken to address the points raised in the previous review. While most responses are justified, one outstanding issue remains: the colocalization of 6Fluo10NAD+ with PAR. While the authors state there is no fixation performed after 6Fluo10NAD staining, it is indicated in the methods in section "Testing Fluorescent NAD On Live Retina" that staining is done on P11, then on P12 tissues are fixed in 4% PFA and processed for imaging. If this is the case, then indeed immunofluorescence localization of PAR as indicated in the literature (e.g. in the work doi.org/10.1021/acs.bioconjchem.9b00089 cited) can be performed simultaneously. It is still not clear why this cannot be performed. While it is appreciated that the authors performed extra experiments trying to image PAR by DAB, the results are unconvincing. PAR staining is performed in the literature with regularity using commercial reagents. The value of 6Fluo10NAD for imaging PARP1 activity must be correlated with PAR staining, otherwise the signal of NAD utilization can be attributed to any number of other enzymes, both in the PARP family and in other families.

7. PLOS authors have the option to publish the peer review history of their article (what does this mean?). If published, this will include your full peer review and any attached files.

Reviewer #2: No

---

## [Author Response · Author response to Decision Letter 1]

21 Dec 2020

We appreciate the reviewer’s thoroughness very much, however, here we think there is a misunderstanding relating to the different types of experiments performed in our study. To assess PARP activity in degenerating retina, we used three different experimental conditions: 1) unfixed (frozen) retinal tissue sections; 2) PFA-fixed retinal tissue sections; and 3) live retinal explant cultures, which were also fixed with PFA. 

6-Fluo-10-NAD+ was first used on unfixed, frozen tissue sections (see the “Histology” section of the Material & Methods for a description of the preparation of the tissue). Since the PARP enzyme cannot “survive” PFA fixation, the PARP activity assay was always conducted on unfixed sections (see the “PARP in situ activity assays” section of the Material & Methods). PARP activity specific signal could readily be revealed by 6-Fluo-10-NAD+ on such unfixed tissue sections (see the Results section “Using NAD+ analogues to probe for in situ PARP activity” and Figure 2).

In addition, 6-Fluo-10-NAD+ was used on live organotypic retinal explants cultures (see the Materials & Methods section “Testing fluorescent NAD+ on live retina”). 6-Fluo-10-NAD+ was added to the culture medium at a concentration of 100µM and for an incubation time of 24h after which the cultures were stopped by PFA-fixation. However, no PARP activity signal could be seen (see Results section “Testing NAD+”). We think that 6-Fluo-10-NAD+ was either not able to penetrate into the tissue or could not reach across the intact cell membranes of live cells (see Discussion section “Tissue penetration and permeability of 6-Fluo-10-NAD+”). 

Taken together, it was not possible to colocalize PARP activity and PAR immunostaining on sections from the organotypic retinal explant cultures, because in that situation our probe did not detect PARP activity. In the previous revision, we provided the result of our trial for colocalizing PARP activity and PAR immunostaining on unfixed dead tissue sections and provided an explanation on why we think it did not work.

To address the reviewer’s question and to further improve the manuscript, we have now inserted an additional comment in the materials and methods, at the beginning of the histology section (page 5, lines 94-96).

---

## [Editor Report · Decision Letter 2]

30 Dec 2020

Fluorescent detection of PARP activity in unfixed tissue

PONE-D-20-16995R2

Dear Dr. Belhadj,

We’re pleased to inform you that your manuscript has been judged scientifically suitable for publication and will be formally accepted for publication once it meets all outstanding technical requirements.

Kind regards,

Alfred S Lewin, Ph.D.

Section Editor

PLOS ONE
---

## [Editor Report · Acceptance letter]

11 Jan 2021

PONE-D-20-16995R2 

Fluorescent detection of PARP activity in unfixed tissue 

Dear Dr. Belhadj:

I'm pleased to inform you that your manuscript has been deemed suitable for publication in PLOS ONE. Congratulations! Your manuscript is now with our production department. 

Kind regards, 

on behalf of

Dr. Alfred S Lewin 

Section Editor

PLOS ONE